# Endocrine-Disrupting Chemicals, Gut Microbiota, and Human (In)Fertility—It Is Time to Consider the Triad

**DOI:** 10.3390/cells11213335

**Published:** 2022-10-22

**Authors:** Gemma Fabozzi, Paola Rebuzzini, Danilo Cimadomo, Mariachiara Allori, Marica Franzago, Liborio Stuppia, Silvia Garagna, Filippo Maria Ubaldi, Maurizio Zuccotti, Laura Rienzi

**Affiliations:** 1B-Woman, Via dei Monti Parioli 6, 00197 Rome, Italy; 2Clinica Valle Giulia, GeneraLife IVF, Via De Notaris 2B, 00197 Rome, Italy; 3Laboratory of Developmental Biology, Department of Biology and Biotechnology “Lazzaro Spallanzani”, University of Pavia, Via Ferrata 9, 27100 Pavia, Italy; 4Center for Advanced Studies and Technology (CAST), University “G. d’Annunzio” of Chieti-Pescara, 66100 Chieti, Italy; 5Department of Medicine and Aging Sciences, University “G. d’Annunzio” of Chieti-Pescara, 66100 Chieti, Italy; 6Department of Psychological, Health and Territorial Sciences, School of Medicine and Health Sciences, University “G. d’Annunzio” of Chieti-Pescara, 66100 Chieti, Italy; 7Centre for Health Technologies (CHT), University of Pavia, Via Ferrata 5, 27100 Pavia, Italy; 8Department of Biomolecular Sciences, University of Urbino “Carlo Bo”, Via Sant’Andrea 34, 61029 Urbino, Italy

**Keywords:** endocrine disruptors, gut microbiota, dysbiosis, reproduction, infertility

## Abstract

The gut microbiota (GM) is a complex and dynamic population of microorganisms living in the human gastrointestinal tract that play an important role in human health and diseases. Recent evidence suggests a strong direct or indirect correlation between GM and both male and female fertility: on the one hand, GM is involved in the regulation of sex hormone levels and in the preservation of the blood–testis barrier integrity; on the other hand, a dysbiotic GM is linked to the onset of pro-inflammatory conditions such as endometriosis or PCOS, which are often associated with infertility. Exposure to endocrine-disrupting chemicals (EDCs) is one of the main causes of GM dysbiosis, with important consequences to the host health and potential transgenerational effects. This perspective article aims to show that the negative effects of EDCs on reproduction are in part due to a dysbiotic GM. We will highlight (i) the link between GM and male and female fertility; (ii) the mechanisms of interaction between EDCs and GM; and (iii) the importance of the maternal–fetal GM axis for offspring growth and development.

## 1. Introduction

In this perspective, we will describe mounting evidence that suggests a damaging effect of endocrine-disrupting chemicals (EDCs) on the gut microbiota (GM), and we will bring experimental, clinical, and epidemiological evidence that shows how dysregulation of the GM eubiotic condition might be the linking ring between EDCs and male and female infertility. Whilst this dysregulation of EDCs on the GM is evident during adult life, we will suggest that it might occur also during the early post-natal phases of life and fetal development, speculating, for the latter, a possible transgenerational effect.

## 2. Gut Microbiota: A New Player in Town

The GM is a complex and dynamic population of microorganisms living in the human gastrointestinal tract that exerts biochemical functions otherwise absent in the host. GM is considered a hidden metabolic organ of the human body influencing human health and diseases [1]. Particularly, GM regulates a range of physiological functions [2] mainly involved in (i) preserving the intestinal barrier integrity [3], (ii) protecting against pathogens [4], (iii) regulating host immunity [5,6], (iv) ensuring energy metabolism [7], and (v) modulating immune development [8].

The main GM phyla are Firmicutes, Bacteroidetes, Actinobacteria, Proteobacteria, Fusobacteria, and Verrucomicrobia, with the former two representing 90% of the whole population [9]. The biodiversity of GM is of utmost importance since it serves as a functional expansion of host genomes. Indeed, the GM’s genome, named gut microbiome, harbors different extra genes encoding enzymatic proteins, non-encoded by the host, that contribute to the regulation of the host physiology [10].

### The Key Role of Gut Microbiota in Health and Disease

The GM is a potential controller of wellness and disease. As demonstrated by experimental, clinical, and epidemiological evidence, a dysbiotic microbiota (i.e., a microbiota that deviates from the “eubiotic” status in terms of diversity and functionality [11]) is implicated in a range of diseases in adults, including inflammatory bowel disease [12,13], arthritis [14], cancer [15], neurological and neuropsychiatric disorders [16], cardiometabolic [17] and cardiovascular [18] disease, obesity [19], type 2 diabetes [20,21,22], and, as detailed below, infertility.

Gut eubiosis is pivotal also for the maintenance of the intestinal barrier integrity, essential to prevent the so-called leaky-gut syndrome (LGS), a condition that causes intestinal permeability resulting in the permeation of antigens, endotoxins, and pathogens and the altered production of neurotransmitters and metabolites such as short-chain fatty acids, leading to chronic low-grade inflammation, a state linked to the development of several diseases [23].

Furthermore, increasing evidence suggests that the acquisition and development of a healthy microbiota in the infant are pivotal to exerting long-lasting beneficial effects in disease prevention [24]. The maternal microbial reservoir is crucial in the maternal-to-infant passage. Maternal vaginal, oral, gut, skin, and breast milk microbial communities contribute to establishing the infant’s own gut microbial community [25,26,27,28], thus regulating correct fetal growth, neurodevelopment, and immune programming [29,30] and providing a prophylactic potential of non-communicable diseases (NCDs), such as obesity, immunoinflammatory disorders, and neurocognitive complications [31].

Extensive microbial colonization takes place post-partum [24] through the mode of delivery, contact with the mother (such as skin-to-skin care), maternal diet, and breastfeeding [32]; however, gut colonization might occur already during the prenatal period [33]. Recent, although controversial [30], findings question the dogma that the womb is sterile, suggesting that the fetus incorporates an initial microbial inoculum already before birth (in utero colonization hypothesis) [29,30,34,35,36], followed by postnatally supplemented maternal microbes.

## 3. The Dangerous Effects of Endocrine-Disrupting Chemicals on the Gut Microbiota

Several factors can affect GM composition and homeostasis, and EDCs are among the most critical [37]. EDCs are highly heterogeneous environmental contaminants that include both natural and synthetic molecules [38]. Phytoestrogens (e.g., genistein and coumestrol) are natural EDCs found in several human and animal food, whereas solvents/lubricants (e.g., polychlorinated biphenyls and dioxins), plastics (e.g., bisphenol A), plasticizers (e.g., phthalates), pesticides and fungicides (e.g., methoxychlor, dichlorodiphenyltrichloroethane, and vinclozolin), and pharmaceutical agents (e.g., diethylstilbestrol) are synthetic EDCs applied to anthropic activities [39]. EDCs enter the organism through the food chain, resulting in adverse interference with hormonally controlled functions [38]. Indeed, they exert their toxicity mimicking estrogen and/or androgen hormone actions, binding to their specific endogenous receptors. They promote impaired activation, synthesis, and secretion of endogenous hormones, thus influencing several hormonal and metabolic processes [38,40]. Moreover, some EDCs show genotoxic effects [41,42] and perturb the epigenetic landscape, inducing alterations of target cells [43,44]. A number of clinical and experimental studies have shown the impact of EDCs on human GM [45]. For instance, increased urinary lead (Pb) was associated with significant changes in the human adult gut microbiota biodiversity [46], affecting both the richness of microbial taxa (α-diversity) and the variability in taxa composition (β-diversity) [47].

Post-natal exposure to di-(2-ethylhexyl) phthalate (DEHP) altered GM composition in new-born babies, with levels of *Bifidobacterium longum*, a key microbial species for normal gut colonization and development [48], significantly decreased compared to control [49]. Moreover, using an in vitro simulator of the human intestinal microbial ecosystem (SHIME), Wang and colleagues demonstrated that BPA exposure significantly changed the variability of the microbial community and increased the percentage of microbes shared in ascending, transverse, and descending colons, observing an upregulated expression of genes related to estrogenic effect and oxidative stress [50].

Many other experimental studies extensively reviewed by Galvez-Ontiveros and colleagues [45] have been conducted in model animals such as rodents, zebrafish, rabbits, and dogs, showing that exposure to polychlorinated biphenyls, parabens, phytoestrogens, metals, triclosan and triclocarban, phthalates, and BPA and its analogs affects GM composition and functionality and triggers metabolic disease.

### Endocrine-Disrupting Chemicals and Gut Microbiota: Mechanisms of Interaction

Gastrointestinal microbiota and EDC interactions are multiple and interdependent. On the one hand, environmental contaminants alter gastrointestinal bacteria composition and/or the metabolic activity that shapes the host’s microbiotype; on the other hand, GM extensively metabolizes environmental chemicals, thus modulating their toxicity in the host [51]. Indeed, the microbiota is pictured as an additional organ involved in xenobiotic biotransformation [52] and influencing the pharmacokinetics of environmental chemicals [53]. Therefore, an altered symbiotic flora may differently modulate chemical toxicity [11,45].

There are three main mechanisms by which EDCs and GM interact [51]:(1)Through the direct and indirect metabolism of xenobiotics, i.e., chemical substances that are not produced by the organism. More specifically, xenobiotics enter the human body mainly through the gastrointestinal tract and reach the distal gut where they can be directly metabolized by GM performing diverse chemical transformations such as hydrolysis, removal of the succinate group, dehydroxylation, acetylation, deacetylation, proteolysis, denitration, deconjugation, or thiazole ring opening. In other circumstances, after ingestion, xenobiotics such as the non-polar ones are transported to the liver for detoxification where they are oxidized and subsequently eliminated in urine or secreted into the bile. In the latter case, they move to the small intestine where they can be absorbed, or they progress down to the large intestine where they are metabolized by the GM.(2)By altering the microbial diversity and thus inducing dysbiosis. For instance, it has been demonstrated that both BPA and ethinylestradiol exposure can elevate the amount of *Bifidobacterium* spp. in mice, leading to metabolic disorders [54]; instead, methylparaben, diethyl phthalate, and triclosan (or their mixture) exposure modifies, in rats, the ratio between *Bacteroidetes* and *Firmicutes* spp. [55], a relevant marker of gut dysbiosis [56]. Moreover, EDCs reduce the number of microbial species such as *Lactobacillus* spp., important for xenobiotic biotransformation and involved in the maintenance of a proper intestinal barrier [57], thus resulting in the enhanced absorption of contaminants and toxicity in the host.(3)EDCs interfere with the GM enzymatic activity. Indeed, GM significantly contributes to the host metabolism by providing enzymes encoded by the gut microbiome (i.e., the genome of GM) which are involved in both the xenobiotic and endobiotic metabolism. Among the most important of these enzymes, β-glycosidase catalyzes the hydrolysis of plant polyphenol glycosides, and β-glucuronidase (GUSB) catalyzes the removal of glucuronic acid from liver-produced glucuronides [58]. Consequently, EDCs, by perturbating GM, may alter host physiological processes mediated by these enzymes.

## 4. Gut Microbiota and (In)Fertility

Emerging evidence indicates that GM composition is key also in reproductive health for both women and men.

### 4.1. Female Fertility

Variations to the GM homeostasis may affect female fertility via different pathways. Primarily, GM is able to influence female fertility by affecting the level of sex hormones [59]. Indeed, as described before, the gut microbiome encodes different enzymes involved in host metabolism, and one of them, the enzyme GUSB, is responsible for the metabolism and modulation of circulating estrogen hormones [60] since it deconjugates estrogens, enabling their binding to estrogen receptors and leading to physiological downstream effects [61,62]. Therefore, changes to the microbial population encoding the enzyme GUSB, known as the estrobolome, affect the endogenous estrogen metabolism by modulating the enterohepatic circulation of these hormones, with a subsequent impact on the woman’s hormonal balance and, therefore, on her fertility [59].

Secondly, GM seems inextricably linked to female infertility due to the important relationship that exists between a healthy GM and the immune system [63]. Indeed, a dysbiotic GM is observed in several infertility-related disorders such as endometriosis [64,65], polycystic ovary syndrome (PCOS), insulin resistance (IR) [66,67,68,69,70,71], and obesity [72,73], characterized by an unbalanced immune profile and pro-inflammatory status, known to negatively affect fertility [74]. All these conditions are characterized by a reduced GM biodiversity and specific microbial imbalances in both the gut and reproductive tract leading to immune dysfunction, compromised immunosurveillance, and altered immune cell profiles. For instance, in women affected by endometriosis, diminished *Lactobacillus* spp. dominance, an altered Firmicutes:Bacteroidetes ratio, and an abundance of vaginosis-related bacteria and other opportunistic pathogens [64,65] determine upregulated ovarian estrogen secretion through neuro-active metabolites that stimulate GnRH neurons, in turn worsening hormonal homeostasis [64]. A further example is that of PCOS patients, in which the GM shows an abnormal *Escherichia:Shigella* ratio and an excess of Bacteroides compared to healthy women [71], a condition occurring also in IR and in obese women characterized by an increased Firmicutes:Bacteroidetes ratio [56,72]. Moreover, GM exerts a role in the pathogenesis of thyroid autoimmune disease, a frequent disorder in infertility patients [75,76,77], and a relationship between the gut microbiome and premature ovarian insufficiency (POI) has been suggested [78,79].

Lastly, GM eubiosis plays a key role in female fertility since it has been demonstrated that GM can influence the whole genital tract microbiota through a continuous crosstalk between uterus and vagina ecosystems [80]. Therefore, a condition of dysbiosis in the gut could possibly lead to vaginal and uterine dysbiosis, negatively affecting endometrial receptivity at the time of implantation [81,82].

Noteworthy, oral administration of probiotics influences vaginal microbiota composition and immunity [23], and different microbial species, such as the Gram-positive *Lactobacillus* spp. that dominates the vaginal microbiota in physiological conditions, originate from the gut [83]. Moreover, GM dysbiosis can induce the leaky-gut syndrome leading to intestinal permeability and leakage of bacteria and bacterial products from the gut into the circulation, thus affecting the female genital tract microbiota [80].

### 4.2. Male Fertility

Growing interest has been devoted also to the role of the microbiome in male reproduction. First, differences in GM between genders have been demonstrated, with a considerably less variety of gut microbiota in men as compared to women, with sex hormones likely responsible for these differences [84,85]. Although the underlying mechanisms by which GM contributes to the regulation of androgens and steroids are still unclear, it is known that some gut microbes can express steroid-processing enzymes and produce steroid hormones with an impact on the metabolism [86,87], and in turn, sex steroids themselves may regulate the structure and function of the GM [87,88].

The GM communicates with the distal organs of the host, including the testis, through various mechanisms, and it may affect male reproduction at several levels.

It has been suggested that a defective gut barrier function can end with the dissemination within the blood flow of microbial-associated molecular patterns (MAMPs) such as lipopolysaccharide, lipoprotein acids, peptidoglycans, and lipoproteins into the testis and epididymis. Indeed, immune system activation, induced by GM translocation, leads to testicular and epididymal inflammation and, also, triggers insulin resistance together with gastrointestinal hormones which, in turn, affect the secretion of sex hormones [luteinizing hormone (LH), follicle-stimulating hormone (FSH), and testosterone (T)] and their role in the regulation of spermatogenesis [89].

This condition determines: (i) the activation of specialized receptors known as pattern recognition receptors which serve as MAMP sensors that recognize the essential microbial components and trigger an immune response; (ii) Langerhans islet inflammation; and (iii) gastrointestinal hormonal changes with insulin resistance and the alteration of LH, FSH, and free T levels [89].

Moreover, a novel role for the GM in the regulation of testicular development and function has been outlined, demonstrating the involvement of the GM in the regulation of the endocrine function of the testis and the integrity of the blood–testis barrier (BTB) [90].

Noteworthy, in addition to the GM, the testicular microenvironment is not completely sterile, containing low amounts of Actinobacteria, Bacteroidetes, Firmicutes, and Proteobacteria. Although the role of testicular bacterial species in the testis remains to be elucidated, it has been suggested that these microbes play an important role in regulating and shaping the immune response in the testis. The gut microbes and testicular microbes together can influence human reproduction, as demonstrated recently in the study of Zhang et al. [91], in which GM altered bile acid levels and affected vitamin A absorption in the gut. Thus, the abnormal vitamin A metabolism can be transferred to the testis through the blood circulation, ultimately resulting in a sharp decline in spermatogonia differentiation in the metabolic syndrome model. Alfano et al. [92] showed a dysbiotic bacterial community in idiopathic nonobstructive azoospermia (iNOA) patients, describing a decrease in Clostridium abundance in those patients with unsuccessful sperm retrieval compared to successful sperm-retrieval patients. This finding has potential clinical relevance since the presence of the class Clostridia has been linked with improved sperm motility and morphology [93,94].

In addition, these authors suggested a link between gut microbes and testicular microbes since testicular bacterial microbiome modifications observed in iNOA men were similar to those previously reported in the gut of elderly individuals [92].

Altered intestinal flora can contribute to increased serum trimethylamine-N-oxide levels, promote vascular inflammation leading to cavernous endothelial and smooth muscle cell damage and to the development of erectile dysfunction [95].

EDCs can cause dysbiosis of the GM with consequent alteration of both the function and the anatomy of the intestinal barrier, activation of the immune system, development of metabolic disorders, and IR which, in turn, affect spermatogenesis and sperm viability [89]. La Merrill et al. [44] identified ten specific key characteristics or main levels where EDCs can interfere with hormone regulation and action, resulting in reproduction disorders. On the other hand, several EDCs compromise BTB integrity and consequently sperm quantity and quality [96], and the GM can protect the germ cells from environmental noxious substances, including EDCs themselves [90]. Indeed, in mice, the GM can modulate the permeability of the blood–testis barrier and influence intratesticular testosterone as well as serum LH and FSH levels. Conversely, the exposure of germ-free mice to commensal bacteria such as *Clostridium tyrobutyricum* can restore the blood–testis barrier integrity [90].

## 5. Endocrine-Disrupting Chemicals, Gut Microbiota, and Reproductive Health: An Intricate Triad with Possible Implication for Future Generations

EDC exposure during adulthood has been clearly associated with negative effect for human health; however, EDC exposure in both pre- and post-natal periods may exert even worse consequences: Firstly, because in this phase the human gut is much less resilient and much more responsive to external and environmental factors than the adult gut. Indeed, fetuses are exposed to a greater risk than adults from food contaminants due to their higher absorption rate, poor detoxification and elimination capacity, faster cell proliferation, and the still immature DNA repair mechanism [36]. Secondly, because during the perinatal and neonatal period, the microbial ecosystem inside undergoes an unprecedented process of shaping [25]. Therefore, any disturbance in this timeframe may lead to more detrimental effects than at any other moment in life affecting both the acquisition and constitution of a healthy GM, with subsequent implications for the exposed individual, offspring health, and their reproductive capability. In light of this, it is not surprising that a growing literature is showing the negative consequences of prenatal EDC exposure on offspring reproductive health.

For instance, it has been recently highlighted that gestational EDC exposure causes serious damage to the reproductive system in male offspring along with disturbing the GM. Gestational exposure to dibutyl phthalate determines an increased abundance of Bacteroidetes, *Prevotella,* and *P. copri* and causes gut dysbacteriosis in the offspring with also an increase in seminiferous atrophy and spermatogenic cell apoptosis [97]. Moreover, phthalate exposure during the gestational period seems to cause testicular damage in the offspring and abnormal phenotypes (e.g., cryptorchidism, loss of reproductive organs, hypospadias) [97]. Lastly, prenatal EDC exposure and the consequent gut–genital microbiota damage are considered at the origin and pathogenesis of endometriosis [65] and male infertility [57,97] in the offspring. Overall, these studies suggest that environmental chemicals may impair in utero programming and, in some cases (e.g., BPA), may be associated with a trans-generationally increased risk of infertility [98].

A key point now is the understanding of the underlying mechanisms regulating the observed transgenerational effects. In this regard, some authors speculated that specific GM strains can induce epigenetic changes in the host genes relevant to immunological, metabolic, and neurological development and functions [24]. For instance, *Lactobacillus* and *Bifidobacterium* can affect DNA methylation by regulating methyl-donor availability through their production of folate [99], or, via butyrate production, they act as histone deacetylase (HDAC) inhibitors and suppressors of nuclear factor-kB (NF-kB) activation, upregulating the expression of PPARγ and decreasing interferon-γ production [100,101]. Moreover, an increased Firmicutes:Bacteroidetes ratio seems to affect DNA methylation [102]. For instance, Kumar and colleagues [103] reported distinct DNA methylation profiles in blood samples from women 6 months after delivery, depending on the predominance of either Firmicutes or Bacteroidetes and Proteobacteria phyla in their fecal microbiota during pregnancy. The differences in methylation patterns affected genes whose function is linked to obesity, metabolism, and inflammation, thus highlighting their link between metabolic disorders and gut microbiota [104].

The exact mechanism by which GM chemically modulate epigenetic marks remains to be clarified. The existence of a “microbiota–nutrient metabolism–host epigenetic” axis has been postulated [105]. According to this hypothesis, the GM acts as a regulator of DNA methylation and histone modifications by altering the levels of nutrients and metabolites: on one hand, directly inhibiting enzymes that catalyze the processes, and on the other, altering the availability of substrates necessary for the enzymatic reactions. In other words, the link between epigenetic marks and GM could be mediated by host-microbial metabolites, acting as substrates and cofactors for epigenetic enzymes. In this scenario, EDCs inducing GM dysbiosis may induce a downstream effect on epigenetic programming and regulation with critical consequences if occurring during the first 1000 days of life of a human individual, a period in which epigenetic DNA imprinting activity is most active and different factors such as nutrition, microbiome, and epigenome play a key role in developmental programming, influencing the individual susceptibility to the development of diseases later in life [24].

## 6. Conclusions and Future Perspectives

Figure 1 highlights the main aspects described in this review: emerging concepts provide evidence of the correlation between EDC exposure, GM dysbiosis, and the occurrence of a range of infertility-related diseases in the exposed individual and in the offspring, shedding light on an intriguing triad (EDCs–GM–(in)fertility) and on its possible involvement in the (dis)regulation of reproductive health in both men and women.

In the era of precision medicine, a better understanding of the role of GM in reproduction opens the possibility to develop novel strategies to prevent or treat infertility and the diseases associated with it, such as maternal dietary modification [106], probiotic and prebiotic supplementation, and fecal microbiota transplantation [107]. Moreover, the recent correlation between microbiota composition and host epigenome suggests that the enrichment for certain microbial species could modulate unique gene expression signatures [8]. The epigenome is a dynamic player in host-microbiota crosstalk functional in precision medicine [108]. Should this fascinating hypothesis be confirmed, the control of epigenetic substrate levels produced by microbial species could represent a new avenue for clinical approaches with pre, pro or post-biotic supplementation to regulate epigenetic enzymes in the gut.

## Figures and Tables

**Figure 1 cells-11-03335-f001:**
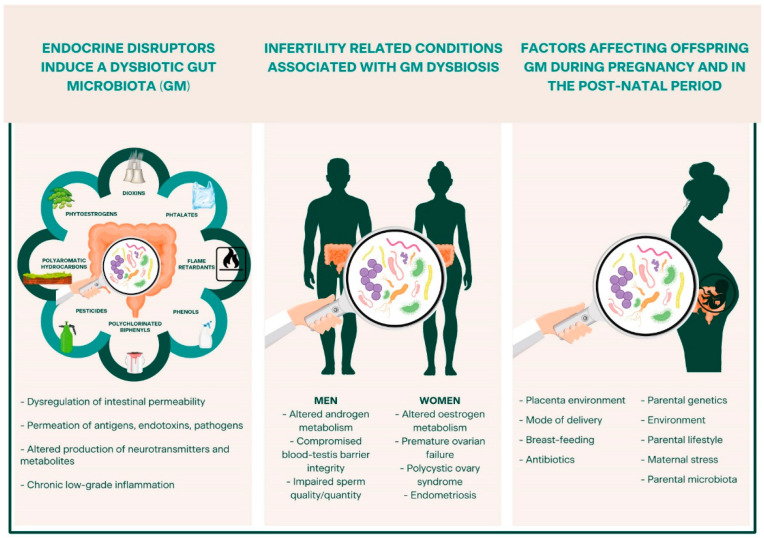
Overview of the intricate relationship among endocrine-disrupting chemicals (EDCs), gut microbiota (GM), and reproductive health. EDCs induce GM damage through several mechanisms of action, leading to GM dysbiosis, intestinal permeability, and chronic low-grade inflammation, conditions linked to several diseases, including those related to infertility. When a dysbiotic condition occurs during pregnancy or in the early post-natal period, it may lead to even worse detrimental effects, affecting both the acquisition and constitution of a healthy GM, with subsequent implications on the exposed individual, the offspring’s health, and reproductive capability.

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
