# Peer review of "Endocrine-Disrupting Chemicals, Gut Microbiota, and Human (In)Fertility—It Is Time to Consider the Triad"

_cells, 2022, doi:10.3390/cells11213335_

Round 1

Reviewer 1 Report

The perspective article by Fabozzi et al describes an exciting and compelling concept of mutual interaction between EDCs and gut microbiota, and its effect on human fertility. I read the paper with great interest and I think so will the journal readers. My only criticism is that the manuscript needs some more thorough editing - it seems that the paragraphs were written by different persons and the editing and English quality is not uniform throughout the text. My detailed comments are as follows:

l. 102 - it should be Pb, not pb.

l. 103 - please, explain what alpha- and beta-diversity is. It is likely that the readers will be predominantly related to the reproduction field, so microbiological terms need some extra explanation.

l. 115-116 - terms like 'Polychlorinated Biphenyls, Parabens, Phytoestro-115 gens, Metals, Triclosan and Triclocarban, Phthalates' should be written with lower case letters.

l. 127 - please, rephrase the sentence, perhaps to 'There are three main mechanisms by which EDCs and GM interact [49]. '

l. 128-129 - awkward style, please, consider rephrasing this sentence.

l.  135 - oxidized, not oxidizes

l. 139 - rephrase the sentence, e.g. to "The second mechanism of action, by which EDCs affect GM, is altering...".

l. 141 - the phrase 'elevate Bifidobacterium' - how can you elevate bacteria? Perhaps you meant elevating their number or amount.

l. 143 - modifies, not modify

l. 143 and in other places in the text - some bacterial latin names are written in italics, some not. I'm not a microbiologist, so perhaps there is some rule I do not know, but please, do double-check it.

l. 144 - "reduce microbial species" - see my comment to l. 141.

l. 146-147 - please, consider rephrasing the sentence to: "A diminished intestinal Lactobacilli dominance results in an enhanced absorption of contaminants and, in turn, toxicity in the host."

l. 150 -  'significantly' would be a better word than 'importantly'

l. 152-153 - awkward style, please, consider rephrasing this sentence.

l. 171 - seems, not seem

l. 175 - it should be '...PCOS), insulin resistance (IR) [64-69], and obesity [70,71].'

l. 181 - in some parts of the text you write 'estrogen', in some (incl. Fig. 1) - 'oestrogen' - please, decide on one spelling.

l. 212 and 218 - It is likely that the readers will be predominantly related to the reproduction field, so immunological terms, such as 'microbial-associated molecular patterns' or 'pattern recognition receptors' need some extra explanation in the text. 

l. 246 - EDCs, not EDs

l. 259 - 'This because, first of all, fetuses are exposed  ....' - consider rephrasing, the style of this sentence is awkward.

l. 262-265 - consider rephrasing, the style of this sentence is awkward.

l. 266 - more detrimental, not worse detrimental

l. 272 - disturbing, not disturbs

l. 307 - it would be good to specify that you meant a human individual here.

Reviewer 2 Report

The authors started from the significance of GM in human health and then gradually expanded to male infertility and female infertility and then further explored the effect from the fetus, which is very well organized! 

In describing the effect of GM on the body and diseases, you mentioned that intestinal laxity only describes the phenomenon, and then mentioned that it is related to the development of many diseases, which should be very important, so the description of him is a bit superficial and does not go deeper into the mechanism of action. When mentioning the effect of EDCs on the composition and homeostasis of GM, it is clearly explained that its mechanism of action is through binding to specific receptors, and then it is mentioned that EDCs shows genotoxic effects leading to changes in the target cells, the examples mentioned are not representative enough, which are used to illustrate its "genotoxic effects". They are not representative, and the use of them to illustrate its "genotoxicity" is untenable.

       The article mentions that the exchange of flora between the uterus and vagina can affect female reproductive health, but it is not clear how the mechanism of action affects female reproductive health, and how the "immune balance" and "inflammatory response" can affect female reproductive health. The "inflammatory response" to premature ovarian failure should be described in more detail, but the article is just a passing comment that leaves a lot to be desired.

In the description of male reproductive effects is very detailed, and in-depth to the basic mechanism of action is very clear, but the use of many clinical cases that do not serve enough, such as iNOA patients with intestinal flora did not mention whether the patient also has other aspects of the disease will have an impact on the flora, should be described in more detail.

The authors also mentioned the "future generations" later, the influence of the state of the infant's flora in utero on the offspring is a very interesting conjecture, and well laid out, most importantly from the genetic aspect is very powerful!
